# Tracking and Vibration Control with a Parallel Structure Controller Based on a Flexible Ball Screw Drive System

**Muzhi Zhu** [1,*], **Dafei Bao** [2], **Mengxin Sun** [1] and **Yong Liu** [1]

1    School of Mechanical Engineering, Nanjing Institute of Technology, Nanjing 211167, China; mxsun@njit.edu.cn (M.S.); ly9824@163.com (Y.L.)
2    Jiangsu Mingzhu Testing Machinery Co., Ltd., Yangzhou 225000, China; baodafei2009@163.com
*    Correspondence: zhumuzhi1021@126.com

**Abstract:** In this paper, we present a parallel structure controller for flexible ball screw drive systems with dynamic variations mainly caused by variations in table position and workpiece mass. The controller consists of two parts: a linear quadratic regulator (LQR) controller with the aim of tracking reference trajectories with high response and accuracy and an interpolated gain-scheduled controller used to restrain system vibration. To damp out the varied resonant modes, the controller is obtained by a set of μ-synthesis linear time-invariant (LTI) controllers interpolated via Youla parameterization. Comparison experiments are conducted to confirm the performance of the proposed controller with a ball screw drive experimental setup. We demonstrate that the parallel structure controller achieves high performance in tracking, vibration suppression and disturbance rejection.

**Keywords:** flexible ball screw drive system; parallel structure controller; gain-scheduled controller; μ synthesis; Youla parameterization

## 1. Introduction

Ball screws are widely used in machine tool feed systems owing to their outstanding advantages like high stiffness, accuracy and transmission efficiency [1]. As machine tools are prevalent equipment for machining metal parts in industry [2], high-speed machining, which can accomplish cost-effective parts manufacturing, is the primary development trend. Therefore, high-performance ball screw drives are required that can operate under high-speed operating conditions. To achieve this goal, two aspects should be considered as prerequisite points. First, the desired cutting trajectory must be followed with fast response and high accuracy for position tracking of the table. Second, the system vibration caused by high-speed operations and external disturbances should be suppressed effectively [3].

Removing the obstacles that affect the high-speed machining performance in tracking and vibration suppression control of flexible ball screw drive systems is difficult. First, under the influence of fast movements with high acceleration, the torsional and axial vibration modes can be excited and produce structural vibrations, which are the definitive factors limiting the achievable bandwidth and deteriorating the accuracy performance of tracking and positioning in a closed-loop system [4]. The system dynamics mainly incited by the table position and workpiece mass variations also act as an important obstacle in high-speed positioning and tracking performance [5,6]. Applied external disturbances like nonlinear friction and cutting force also hinder the trajectory-tracking accuracy [7]. External disturbances applied near resonant frequencies can also cause structural vibrations in the machine. Therefore, to satisfy the requirements of high-speed machining, the mentioned obstacles should be removed in the design of ball screw drive controllers.

With classical controllers, the demands of high-precision control with tracking and vibration suppression cannot be realized [8]. In recent decades, various tracking and vibration control methods for ball screw drive systems have been proposed. Erkorkmaz and Kamalzadeh [9] designed a sliding-mode controller for tracking control and a notch

filtering in a rigid model of a ball screw drive system to filter out the first-order vibration. Combined with pole-placement principle, they further developed an adaptive sliding-mode controller with a two-degrees-of-freedom model, which has a substantial improvement in terms of both vibration control and tracking performance [10]. Okwudire and Altintas [11] proposed a robust discrete-time sliding-mode controller to compensate for the vibration modes by considering the structural flexibility and disturbance effect. Considering the time-varying parametric uncertainties of flexible ball screw drive systems, Dong [12] proposed an adaptive backstepping sliding-mode controller that improved the tracking accuracy. The pole-placement technique was employed by Gordon and Erkorkmaz [13] to compensate for active structural vibrations and then achieve high bandwidth and positioning accuracy. Altintas and Khoshdarregir [14] eliminated residual vibrations in a CNC machine by applying input-shaping filters to the reference commands. Fujimoto [15] proposed a repetitive perfect tracking controller with an n-times learning filter for a ball screw drive system to achieve high precision. To avoid excitation of resonant modes in high-precision machining, a learning controller was employed by Tsai et al. [16] to filter out undesired signals from reference commands. Zhang [17] used disturbance-rejection control and equivalent error-model-based feedforward control to ensure tracking performance and compensate for the uncertainties of a ball screw drive system. Rajabi et al. [18] proposed trajectory-tracking control of a ball-screw-driven servomechanism based on a sliding-mode approach with state estimation by an extended/unscented Kalman filter. A predefined performance-constrained, non-singular sliding-mode control was proposed by Park et al. [19], in which the assumed feedforward dynamic method and super-twisting state observer are combined to compensate for unknown dynamics and unmeasured velocity signals in ball screw drive systems. Sun et al. [20] presented a cascade-structured controller consisting of a weakly set motor speed controller, a disturbance observer with the feedback velocity of ball screw drives and a PD position controller. For precise control of ball screw drives, Hayashi et al. [21,22] used projection-based iterative learning control to deal with both variation in position reference and rolling friction compensation. Sencer and Dumanli [23,24] presented an optimal non-collocated control stage for flexible ball screw feed drives designed with only load (table)-side feedback signals to achieve high tracking bandwidth, disturbance rejection and modal damping. Yang et al. [25] proposed a dual-position feedback control method by introducing the drive-side position information to the position loop feedback channel with a filter and an adaptive backlash error compensation method, which can reduce the over-quadrant error. Recently, Shirvani et al. [26] used adaptive feedforward cancellation to address the problem of harmonic positioning error suppression in ball screw drives.

The controller design mentioned above does not explicitly focus on dynamic time-variant characteristics of the ball screw drives. The gain-scheduled method is capable of handling a linear parameter-varying (LPV) system. Considering the stiffness variations of ball screw drives, Symens [27] first proposed the gain-scheduled H∞ controller. Then, Silva [28] and Paijmans [29] designed an LPV gain-scheduled controller using an interpolating technique to realize system-varying dynamic compensation.

Previous works only aimed at a single-input–single-output (SISO) system with single scheduling parameter situation. Sepasi [30] synthesized a robust gain-scheduling con-troller for tracking control of ball screw drive systems to cope with the dynamic variations of a multi-input–multi-output (MIMO) system, but external disturbances were not considered in the study. Dong et al. [31] extended an interpolation-based approach to the design of a gain-scheduled H∞ loop-shaping controller for ball screw drives with table position and workpiece mass as the two scheduling parameters; however, they did not consider the effect of the cutting force disturbances. Hanifzadegan [32] proposed a switching gain-scheduled controller with an extensive range of applications using an analytical LPV model of flexible ball screws. He also [33] proposed a parallel structure feedback controller that contained a robust LTI tracking controller and a robust switching gain-scheduled LPV controller for vibration suppression. Recently, Zhang et al. [34] presented an LPV-model-based

gain-scheduling control method considering the time-varying and rigid–flexible coupling dynamic characteristics of ball screw feed drives. Nevertheless, the LPV gain-scheduled approach has a complex structure, and the solving process is difficult.

Above all, the controller designed for LTI models cannot explicitly deal with dynamic variation of ball screw drives, such as sliding-mode control, pole-placement control, positive position feedback control, optimal non-collocated control, etc., as mentioned previously. In addition, a gain-scheduling controller can cover a large range of parameter variations in vibration suppression, but transient behavior of the controller at switching instants might degrade the tracking performance. The goal of this paper is to propose a controller structure suitable for ball screw drives with significant dynamic variations in order to achieve high tracking and vibration suppression performance. A parallel structure controller composed of two controllers is adopted to obtain the respective objectives: one is an LTI linear quadratic regulator (LQR) controller with full-state feedback, which takes charge of improving tracking accuracy and speed for reference trajectories, whereas the other is an interpolating gain-scheduled controller used to damp out the flexible modes with varying resonant frequencies according to the table position and the workpiece mass variations. During the control process, several local μ-synthesis controllers are adopted first to suppress the varying flexible modes in each table position subregion and load masses. Then, the gain-scheduled controller can be obtained by dynamic transition between μ-synthesis controllers, which is termed controller interpolation. Because switching robust controllers may induce undesirable transients and lead to performance losses and instability [35–37], Youla controller parameterization is used to ensure transient performance and system stability. The comparative experiments are also designed to verify the effectiveness of the overall control strategy in comparison with a proportional–integral (P-PI) controller.

The content of this paper is organized as follows. The LPV model for a flexible ball screw drive system is presented in Section 2. In Section 3, a tracking controller based on LQR is designed. In Section 4, we present an interpolating gain-scheduled μ-synthesis controller designed for structural vibration suppression. The experimental setup and results are presented in Section 5, and conclusions are drawn in Section 6.

## 2. LPV Model for a Flexible Ball Screw Drive System

The fundamental ball screw drive system is powered by a servomotor by applying a voltage; then, a torque is generated to drive the screw, which is connected to the motor. The screw rotational motion is translated to the table linear motion actuated by the nut. An external disturbance such as a cutting force or friction is applied onto the table under actual working conditions. Therefore, the flexible ball screw drive system can be modeled as a two-input–two-output system. Because the focus of this paper is on the compensation of the first-order vibration mode, the transfer functions ($G_L(m, l_t)$) between inputs and outputs can be expressed as

$$\begin{bmatrix} l(s) \\ \theta(s) \end{bmatrix} = G_L(m, l_t) \begin{bmatrix} u(s) \\ d(s) \end{bmatrix} \tag{1}$$

$$G_L(m, l_t) = \frac{R_L}{s(js + b)} \times \frac{P_L(m, l_t) + Q_L(m, l_t)s}{s^2 + 2\zeta\omega(m, l_t)s + \omega^2(m, l_t)} \tag{2}$$

where the inputs are the command voltage ($u$) and external disturbances ($d$), whereas the outputs are the rotating displacement of the motor ($\theta$) and the linear displacement of the table ($l$). The table position ($l_t$) is time-varying during the working process, whereas the workpiece mass ($m$) is assumed to be constant in this study. $m$ and $l_t$ are the gain-scheduling variables of the LPV model. The parameter $j$ represents the equivalent inertial of the rigid body mode, and parameter $b$ represents the damping. $\omega$ and $\zeta$ are the natural frequency and damping ratio of the first flexible mode. $P_L$, $Q_L$ and $R_L$ are the $2 \times 2$ matrices that determine the gain and zeros of transfer functions.

The transfer functions of ball screw drives can be converted to a two-degrees-of-freedom system as shown in Figure 1. The state space function of the system can be written as

$$
\begin{cases}
\dot{x}(t) = A_L(m, l_t) x(t) + B_L(m, l_t) \begin{bmatrix} u(t) \\ d(t) \end{bmatrix} \\
\\
\quad\quad \begin{bmatrix} l(t) \\ \theta(t) \end{bmatrix} = C_L x(t)
\end{cases}
\tag{3}
$$

$$
A_L(m, l_t) =
\begin{bmatrix}
0 & 0 & 1 & 0 \\
0 & 0 & 0 & 1 \\
-\dfrac{k(m, l_t)}{m_2(m)} & \dfrac{k(m, l_t)}{m_2(m)} & -\dfrac{b_2 + c(m, l_t)}{m_2(m)} & \dfrac{c(m, l_t)}{m_2(m)} \\
\dfrac{k(m, l_t)}{m_1(m)} & -\dfrac{k(m, l_t)}{m_1(m)} & \dfrac{c(m, l_t)}{m_1(m)} & -\dfrac{b_1 + c(m, l_t)}{m_1(m)}
\end{bmatrix}
\tag{4}
$$

$$
B_L(m, l_t) =
\begin{bmatrix}
0 & 0 \\
0 & 0 \\
0 & \dfrac{1}{m_2(m)} \\
\dfrac{1}{m_1(m)} & 0
\end{bmatrix},
\quad
C_L(m, l_t) =
\begin{bmatrix}
1 & 0 & 0 & 0 \\
0 & 1 & 0 & 0
\end{bmatrix}
$$

where $x(t) = \begin{bmatrix} x_2 & x_1 & \dot{x}_2 & \dot{x}_1 \end{bmatrix}^T$ is the state vector, and $x_1$ and $x_2$ represent the motor rotational displacement and table displacement, respectively. $k(m, l_t)$ is the overall axial stiffness coefficient affected by the preloaded nut, thrust bearing and screw. $c(m, l_t)$ is the damping coefficient induced by the preloaded nut. $k(m, l_t)$ and $c(m, l_t)$ are continuous time-varying functions that are significantly dependent on table position ($l_t$) and workpiece mass ($m$). $m_1(m)$ and $m_2(m)$ represent the equivalent inertia of rotating parts and the equivalent inertia of the table, respectively, and are functions only subjected to the workpiece mass ($m$). $b_1$ is the viscous damping coefficient of rotating parts, such as the motor and bearings, and $b_2$ is the viscous damping coefficient of the guideways.

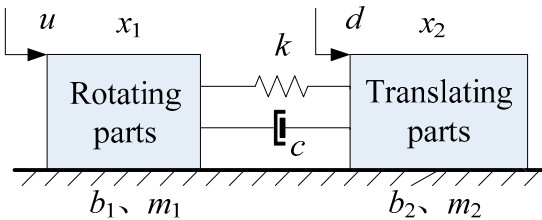

**Figure 1.** Two-degrees-of-freedom model.

## 3. Parallel Controller Structure

In this paper, we separate the multi-objective control problem (tracking and vibration suppression) into two subproblems. (1) One controller is responsible for accurate reference trajectory tracking in the low-frequency range, and (2) another controller is responsible for vibration suppression near the resonant frequencies.

The parallel structure of controllers is shown in Figure 2. As previously mentioned, it consists of two controllers: a tracking controller ($K_{\text{Track}}$) and a vibration suppression controller ($K_{\text{Vib}}$). The $K_{\text{Track}}$ controller is used to increase the closed-loop bandwidth and minimize the tracking error. On the other hand, the vibration suppression controller ($K_{\text{Vib}}$) is used to reduce the structural vibration caused by external disturbances ($d$) by feeding back the table linear displacement ($l$) and the motor rotating displacement ($\theta$). Two controller outputs ($u_{\text{Track}}$ and $u_{\text{Vib}}$) are summed and applied to the servo motor.

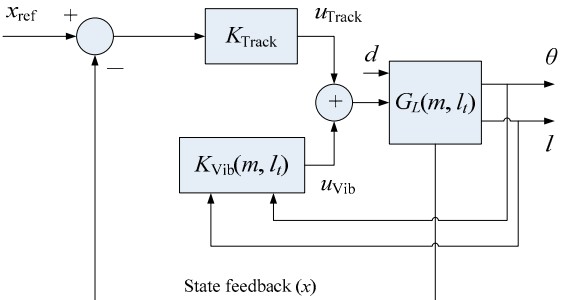

**Figure 2.** Block diagram of a parallel controller structure.

Compared to MIMO controllers without any specific structure, the parallel structure of controller shown in Figure 2 has several advantages. (1) For most multi-objective control problems, the designed controller parameters need to be iteratively tuned to balance the different control objectives. In contrast, for the parallel structure controller, the parameters of the two controllers can be adjusted separately to improve the corresponding control performance without considering other parameters that influence other performance aspects, which simplifies the controller design. (2) The unduly high degrees generated by a general MIMO controller introduce considerable computational pressures in the microcontroller. However, the designed controller consists of two relatively low-degree controllers that reduce computational burdens in its implementation with a microcontroller. (3) In the manufacturing industry, servo drives with tracking controllers are already common. Therefore, adding a vibration controller to an existing tracking controller can reduce structural vibration without reducing tracking performance if using a parallel structure.

## 4. Tracking Controller Design Based on LQR

To ensure tracking performance, a full-state feedback tracking controller based on LQR is designed. Because the frequencies of reference signal are much lower than first axial resonant modes, the state space function (3) can be simplified as a nominal model by ignoring the external disturbance and dynamic variations. The one-input–two-output nominal model is expressed as

$$\begin{cases} \dot{x}(t) = A_N x(t) + B_N u_{\text{Track}}(t) \\ \begin{bmatrix} l(t) \\ \theta(t) \end{bmatrix} = C_N x(t) \end{cases} \tag{5}$$

$$A_N = \begin{bmatrix} 0 & 0 & 1 & 0 \\ 0 & 0 & 0 & 1 \\ -\dfrac{k_N}{m_{2N}} & \dfrac{k_N}{m_{2N}} & -\dfrac{b_2 + c_N}{m_{2N}} & \dfrac{c_N}{m_{2N}} \\ \dfrac{k_N}{m_{1N}} & -\dfrac{k_N}{m_{1N}} & \dfrac{c_N}{m_{1N}} & -\dfrac{b_1 + c_N}{m_{1N}} \end{bmatrix}, \quad B_N = \begin{bmatrix} 0 & 0 \\ 0 & 0 \\ 0 & \dfrac{1}{m_{2N}} \\ \dfrac{1}{m_{1N}} & 0 \end{bmatrix}, \tag{6}$$

$$C_N = \begin{bmatrix} 1 & 0 & 0 & 0 \\ 0 & 1 & 0 & 0 \end{bmatrix}$$

The feedback structure with an LQR controller is depicted in Figure 3, where $x_{ref}$ is the reference signal, which contains the desired displacements and velocities of the table and motor. In order to increase the closed-loop bandwidth and minimize the tracking error, the cost function ($J$) should be minimized as follows:

$$J = \frac{1}{2} \int_0^\infty \left( \left( x_{ref} - x \right)^{\text{T}} Q \left( x_{ref} - x \right) + u^{\text{T}} R u \right) \mathrm{d}t \tag{7}$$

where $Q \geq 0$ and $R > 0$ are $4 \times 4$ and $1 \times 1$ symmetric, positive (semi-)definite matrices, respectively, that can be chosen suitably to achieve the desired control effect. The proposed LQR controller design can be solved by the MATLAB command lqr($A_N$, $B_N$, $Q$, $R$).

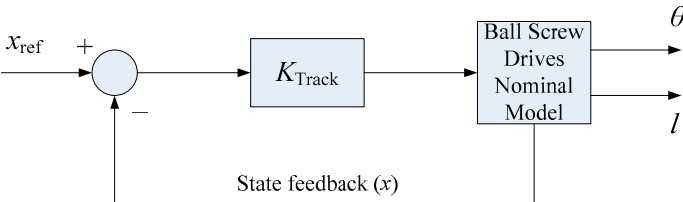

**Figure 3.** Tracking controller structure.

## 5. Structural Vibration Controller Design Based on Interpolating Gain-Scheduled µ Synthesis

### 5.1. µ-Synthesis Design

The stiffness and damping coefficient of a ball screw drive system are varied depending on the table location and workpiece mass, whereas the table location is time-varying during machining. With the varying stiffness and damping coefficient, the ball screw system has dynamic vibration modes. To suppress the dynamic vibration modes, a parallel controller ($K_{\text{Vib}}(m, l_t)$) that is gain-scheduled by $l_t$ and $m$ is designed. First, a set of LTI controllers is obtained through the µ-synthesis method to ensure the robustness of the system within a certain variation range of $l_t$. As shown in Figure 1, because the translating parts are affected by external disturbances ($d$), they may induce vibration near the resonance frequencies of the ball screw drive. Thus, the objective of the µ-synthesis controller is to evaluate the control signal ($u_\mu$) in order to reduce the oscillations of the translating parts due to external disturbances. The LTI control model ($G_{\text{Vib}}$) is integrated in the ball screw drive system LTI model with stiffness and damping coefficient uncertainties caused by position variation and the tracking controller ($K_{\text{Track}}$), as expressed by

$$\begin{cases} \dot{x}(t) = (A_{\text{LTI}} - B_{1N}K_{\text{Track}})x(t) + B_N \begin{bmatrix} u_\mu(t) \\ d(t) \end{bmatrix} \\ \begin{bmatrix} l(t) \\ \theta(t) \end{bmatrix} = C_N x(t) \end{cases} \tag{8}$$

$$A_{\text{LTI}} = \begin{bmatrix} 0 & 0 & 1 & 0 \\ 0 & 0 & 0 & 1 \\ -\dfrac{\bar{k}_N}{m_{2N}} & \dfrac{\bar{k}_N}{m_{2N}} & -\dfrac{b_2 + \bar{c}_N}{m_{2N}} & \dfrac{\bar{c}_N}{m_{2N}} \\ \dfrac{\bar{k}_N}{m_{1N}} & -\dfrac{\bar{k}_N}{m_{1N}} & \dfrac{c_N}{m_{1N}} & -\dfrac{b_1 + \bar{c}_N}{m_{1N}} \end{bmatrix} \tag{9}$$

where the values of stiffness ($\bar{k}_N$) and the damping coefficient ($\bar{c}_N$) are uncertain, with nominal values of $k_N$ and $c_N$, respectively, and can vary within a given range depending on the variation of $l_t$. $B_{1N}$ is the first column of the $B_N$ matrix. Then, the LTI control plant ($G_{\text{Vib}}$) with parameter uncertainties can be derived as

$$\begin{bmatrix} \dot{\tilde{x}} \\ \cdots \\ z \\ \cdots \\ y \end{bmatrix} = \begin{bmatrix} \tilde{A} & \vdots & \tilde{B}_1 & \vdots & \tilde{B}_2 \\ \cdots & \cdots & \cdots & \cdots & \cdots \\ \tilde{C}_1 & \vdots & \tilde{D}_{11} & \vdots & \tilde{D}_{12} \\ \cdots & \vdots & \cdots & \vdots & \cdots \\ \tilde{C}_2 & \vdots & \tilde{D}_{21} & \vdots & \tilde{D}_{22} \end{bmatrix} = \begin{bmatrix} \tilde{x} \\ \cdots \\ w \\ \cdots \\ \tilde{u} \end{bmatrix} \tag{10}$$

where $\widetilde{x}$ is the total state vector, $w = [w_k \; w_c]^{\mathrm{T}}$ is defined as the input uncertainties associated with the stiffness $(\bar{k}_N)$ and damping coefficient $\bar{c}_N$ and $z = [z_k \; z_c]^{\mathrm{T}}$ is the output uncertainties. $y = [l \; \theta]^{\mathrm{T}}$, $\widetilde{u} = [u_\mu \; d]^{\mathrm{T}}$ and the system matrices are given in Appendix A. The feedback structure with a μ-synthesis controller is shown in Figure 4, where $W_p$ represents the frequency performance weight function, which is used to damp out the vibration amplitude. $W_u$ is defined as the control weight function to penalize the control signal $(u_\mu)$ at the specified frequency ranges due to external disturbance $(d)$. $W_{\mathrm{dist}}$ is the disturbance performance weight function, which is used to compensate for the effect of disturbances. Then, the open-loop interconnected transfer function matrix $(P_{\mathrm{LTI}})$ shown in Figure 4 can be partitioned as Equation (11).

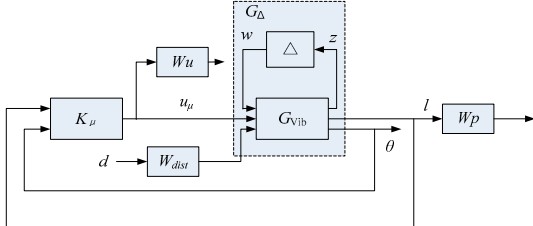

**Figure 4.** The structure of a μ-synthesis controller.

$$P_{\mathrm{LTI}} = \begin{pmatrix} P_{11} & P_{12} & P_{13} \\ P_{21} & P_{22} & P_{23} \\ P_{31} & P_{32} & P_{33} \end{pmatrix} = \left( \begin{array}{c|ccc|c} G_{\mathrm{Vib}} & 0 & 0 & 0 \\ \hline W_p G_{\mathrm{Vib}} & W_p & W_p W_{\mathrm{dist}} & W_p G_{\mathrm{Vib}} \\ 0 & 0 & 0 & W_u \\ \hline G_{\mathrm{Vib}} & I & W_{\mathrm{dist}} & G_{\mathrm{Vib}} \end{array} \right) \tag{11}$$

The block matrix $(\Delta_P)$ is defined as

$$\Delta_P = \left\{ \begin{bmatrix} \Delta & 0 \\ 0 & \Delta_F \end{bmatrix} : \Delta \in R^{2\times2}, \Delta_F \in C^{1\times2} \right\} \tag{12}$$

where the $\Delta$ is diagonal matrix and corresponds to the uncertainties of stiffness and the damping coefficient in the model. Another block $(\Delta_F)$ represents the fictitious uncertainty used to achieve the performance requirements of the $\mu$ approach.

The aim of the controller design is to obtain a stabilizing $K_\mu$ so that for each frequency $(\omega \in [0, \infty])$ the structured singular value satisfies the following condition [38]:

$$\mu_{\Delta_P}\left[F_L\left(P_{\mathrm{LTI}}, K_\mu\right)(j\omega)\right] < \gamma \tag{13}$$

where $F_L\left(P_{\mathrm{LTI}}, K_\mu\right)$ is the lower linear fractional transformation of $P_{\mathrm{LTI}}$ and $K_\mu$ and $\gamma$ is normalized to 1. The procedure described above contains all the essential ingredients of $\mu$ synthesis required for an LTI controller $(K_\mu)$ to be obtained.

### 5.2. Interpolating Gain-Scheduled Controller Design via Youla Parameterization

An interpolating gain-scheduled vibration controller is composed of a supervisor controller and the corresponding interpolated controller, as shown in Figure 5. The supervisory controller generates the interpolation signals $(\alpha(t)$ and $\beta)$, which are assigned according to the two scheduling parameters of table position $(l_t)$ and workpiece mass $(m)$, respectively. Interpolation signals describe which fraction of each local controller is active within the interpolated controller. $K_{\mathrm{Vib}}(\alpha, \beta)$ is the interpolated controller, which comprises of a set of local controllers $(K_{\mu ij}, i = 1, \ldots, n. \; j = 1, \ldots, m)$. As switching among local μ-synthesis controllers may lead to instability, the Youla parameters are applied to maintain the system stability with gradual switching. The authors of [39] reported that any robust controller $(K)$ that internally stabilizes a plant $(P)$ can be written as a linear fractional transform (LFT):

$$K(Q) = \mathrm{LFT}(J, Q) \tag{14}$$

where the fixed subsystem ($J$) and the exponentially stable Youla parameter ($Q \in RH_\infty$) can expressed as

$$J = \begin{cases} \dot{\hat{x}} = A_\mu \hat{x} + Z_\mu P_{31}^T y + (I + \gamma^{-2} Z_\mu X_\mu) P_{13} \hat{u} \\ u = F_\mu \hat{x} + \hat{u} \\ \hat{y} = -P_{31} \hat{x} + y \end{cases} \tag{15}$$

$$Q = \begin{cases} \dot{x}_q = A_q x_q + B_q \hat{y} \\ \hat{u} = C_q x_q + D_q \hat{y} \end{cases}. \tag{16}$$

The $A_\mu$, $F_\mu$, $X_\mu$ and $Z_\mu$ matrices are defined in Appendix B.

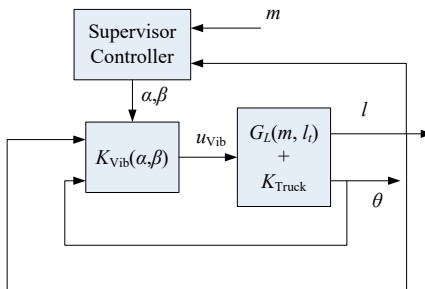

**Figure 5.** The structure of an interpolating gain-scheduled vibration controller.

According to this theory, several μ controllers are allowed to switch with each other, and the system stability is guaranteed by the fact that a convex combination of stable transfer matrices ($Q_{ij}$) corresponding to $K_{\mu ij}$ is always stable. Because the workpiece mass ($m$) is constant during the working process, the interpolation signals ($\beta_j$) are defined as discrete signals, i.e., $\beta_j = 0, 1, j = 1, \ldots, m$. For the interpolation signals ($\alpha_j(t)$), a smooth transition during the switching phase is desirable. Given that the table position ($l_t$) is continuously variable, two functional blocks ($F_{1j}$ and $F_{2j}$) are established; block $F_{1j}$ stores the odd-numbered Youla parameters ($Q_{1j}, Q_{3j}, \ldots, Q_{(i-1)j}, i = 1, \ldots, n$), and block $F_{2j}$ contains even-numbered Youla parameters ($Q_{2j}, Q_{4j}, \ldots, Q_{ij}$). Then, a smooth trajectory of the interpolation signals ($\alpha_j(t) \in [0, 1]$) is applied. The supervisory control law is demonstrated in Figure 6, which shows that block $F_{1j}$ is active when $\alpha_j(t) = 0$, whereas $\alpha_j(t) = 1$ enables block $F_{2j}$. When the table position ($l_t$) passes through the switch point ($l_{i,i+1}$), the controller enters into the switching phase from $F_{1j}$ to $F_{2j}$ or, conversely, $\alpha_j(t) \in [0, 1]$ changes from 0 to 1 or the inverse in the specified time interval. Then, the interconnection of the gain-scheduling interpolated controller ($K_{\text{Vib}}(\alpha, \beta)$) can be expressed as shown in Figure 7. According to the Figure 6, the corresponding gain-scheduling Youla parameter ($Q$) can be written as

$$Q = \sum_{j=1}^{m} \beta_j ((1 - \alpha_j) F_{1j} + \alpha_j F_{2j}) \tag{17}$$

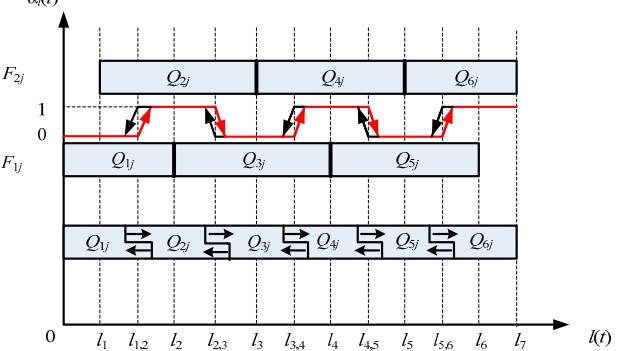

**Figure 6.** Supervisory control law of $\alpha i(t)$ for interpolation.

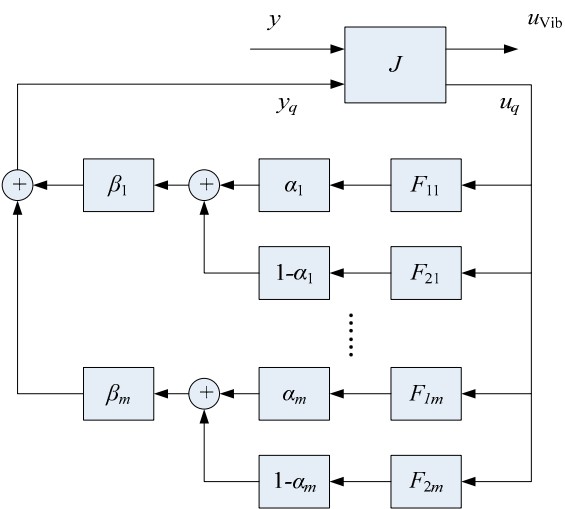

**Figure 7.** The interconnection of a gain-scheduling interpolated controller.

## 6. Experimental Results

### 6.1. Experimental Setup

The ball screw drive experimental setup is powered by a 9 kW Mitsubishi servo motor and driven by an NSK ball screw with 20 mm pitch and a 20 mm diameter. A 50 kg working table travels in the range of 0 to 560 mm. A Heidenhain incremental rotary encoder is available on the motor side of the ball screw, which delivers 5000 P/Rev and can be reliably interpolated 25 times in the Heidenhain IBV660 interpolation and digitizing electronics, resulting in a position measurement resolution equivalent to 0.04 μm. A Heidenhain incremental linear encoder is installed on the table with a 5 μm signal period, which is interpolated by 100 times with a 0.05 μm position measurement resolution. These devices are used to communicate position and velocity feedback signals of both the table and the screw. The dSPACE DS1103 controller is used as a development system for rapid control prototyping. The structure of the experimental setup is shown in Figure 8.

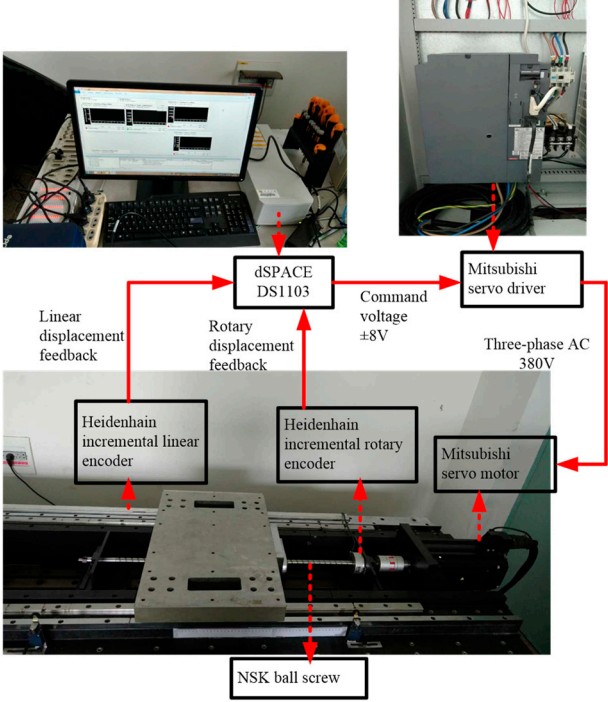

**Figure 8.** Ball screw drive experimental setup.

### 6.2. System Identification

Several two-degrees-of-freedom nominal models are identified. The parameters of rigid body mode, including inertia (*j*) and viscous damping (*b*) were identified using least squares identification techniques.

The length range of the ball screw was fixed at $l \in [50,380]$ mm and divided into eight intervals: [50,100] mm, [90,140] mm, [130,180] mm, [170,220] mm, [210,260] mm, [250,300] mm, [290,340] mm and [330,380] mm. In order to identify the parameters of the flexible-mode ball screw drive experimental setup, frequency response function (FRF) measurements were performed by placing the table at the midpoint of the eight intervals with the workpiece mass (*m*) equal to 0 kg and 25 kg. The frequency range of the sweeping signal was from 2 Hz to 400 Hz with an increment of 2 Hz. The amplitudes and phases at each frequency were calculated using least squares techniques. The frequency response functions were converted into continuous time transfer functions. According to the parameters of the rigid body mode and flexible mode, the parameters of two-degrees-of-freedom nominal models can be obtained by solving the overdetermined equations with the least squares method. The FRF of the ball screw drive experimental setup at each of the eight intervals with a 0 kg workpiece mass is shown in Figure 9, with the control voltage (*u*) and table position (*l*) as the input and output, respectively.

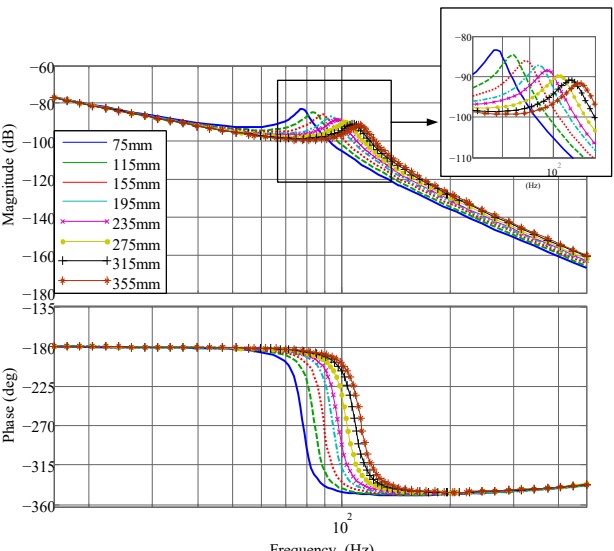

**Figure 9.** The FRFs of the ball screw drive experimental setup at eight intervals with a 0 kg workpiece mass.

### 6.3. Controller Design for Experimental Setup

First, the tracking controller ($K_{\text{Track}}$) for the nominal model of the experimental setup is designed. The parameters of the nominal model are listed in Table 1. *Q* and *R* are selected suitably to improve the tracking performance of the system. *Q* is chosen as a diagonal matrix, where $q_1$, $q_2$, $q_3$ and $q_4$ represent the effort made by each state variable to theoretically reach the reference value. Similarly, *R* is used to adjust the balance between control voltage ($u_{\text{Track}}$) and states. Finally, adjusting parameters through simulation and experimentation is indispensable. Then, the parameters of weighting matrices *Q* and *R* are selected as

$$Q = \begin{bmatrix} 320,000 & & & \\ & 280,000 & & \\ & & 0.00004 & \\ & & & 0.00003 \end{bmatrix}, \ R = 0.00003 \tag{18}$$

**Table 1.** Parameters of the ball screw drive experimental setup nominal model.

| Symbol | Value | Unit |
| --- | --- | --- |
| $m_1$ | 0.6512 | $(\mathrm{V \cdot s^2 \cdot m^{-1}})$ |
| $m_2$ | 0.0771 | $(\mathrm{V \cdot s^2 \cdot m^{-1}})$ |
| $b_1$ | $4.1571 \times 10^{-4}$ | $(\mathrm{V \cdot s \cdot m^{-1}})$ |
| $b_2$ | 0.8052 | $(\mathrm{V \cdot s \cdot m^{-1}})$ |
| $k$ | $2.1153 \times 10^{-4}$ | $(\mathrm{V \cdot m^{-1}})$ |
| $c$ | 2.6775 | $(\mathrm{V \cdot s \cdot m^{-1}})$ |

Second, the gain-scheduled structural vibration controller ($K_{\mathrm{Vib}}$) is synthesized. The first step is to design distinct $8 \times 2$ LTI $\mu$-synthesis controllers for $8 \times 2$ LTI control models with uncertainties. The simulated FRF of the LTI control model combined with the LTI experimental setup model with uncertainties in the position of $l \in [170,220]$ mm with a tracking controller ($K_{\mathrm{Track}}$) is displayed in Figure 10, with the control voltage ($u_{vib}$) and the table position ($l$) as input and output, respectively. $W_{\mathrm{dist}}$, $W_p$ and $W_u$ are selected suitably to improve disturbance suppression and vibration suppression by damping out resonant modes in local positions. Low-pass filters can be selected as $W_p$ to shape the sensitivity function (S). To improve the vibration suppression by damping out resonant modes, high-frequency gain can be decreased and low-frequency can be increased of $W_p$, and the crossing frequency can be set below the first resonant frequency of each position ($l$) and mass ($m$) interval, an appropriate degree searched from low to high. For the control weighting function ($W_u$), a constant can be used to limit the input voltage ($u_{\mathrm{Vib}}$) and avoid motor saturation. $W_{\mathrm{dist}}$ is usually selected based on the frequency domain characteristics of the disturbance. Frequency domain analysis of the effect of machine tool cutting forces on feed drive control is needed in real cutting processes. The next step is to synthesize a structural vibration controller ($K_{\mathrm{Vib}}$) based on the $8 \times 2$ $\mu$-synthesis controllers according to Section 5.2.

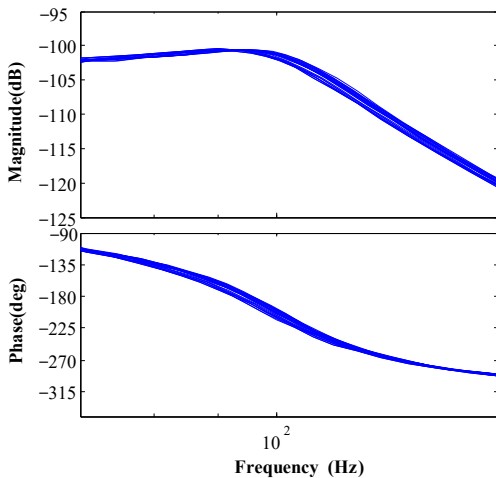

**Figure 10.** The FRF of the LTI control model in a position of $l \in [170,220]$ mm with a tracking controller ($K_{\mathrm{Track}}$).

*6.4. Tracking Performance Experiments*

With a displacement of 330 mm, a jerk continuous trajectory is considered as the reference signal. The maximum speed is 0.25 m/s, and the maximum acceleration is 0.25 g m/s², as shown in Figure 11. As a comparison, the $K_{\mathrm{Track}}$ controller and the P-PI controller with velocity feedforward are applied in the performance experiments. The purpose of using the $K_{\mathrm{Track}}$ controller alone is to investigate the influence of $K_{\mathrm{Vib}}$ controllers in parallel structures. Otherwise, a P-PI controller is chosen for comparison because it is widely used in the motion control industry. In the P-PI controller, the velocity loop adopts PI (proportional–integral) control, in which the input is the velocity error. The position loop

relies on proportional control, in which the input is the position-tracking error. The output of the P-PI controller is the motor voltage. To improve the tracking performance, velocity feedforward is added to the P-PI controller with a velocity reference signal. The velocity-proportional, integral and position-proportional gains are selected as 1000, 200 and 8000, respectively, and the velocity feedforward coefficient is tuned as 1. Each test was conducted three times in order to confirm the reliability and repeatability of the experimental results and observe a consistent performance. The experimental results provide a comparison of the performance in cases 1 and case 3 with that in case 2.

- Case 1: only a $K_{Track}$ controller;
- Case 2: both a $K_{Track}$ controller and a $K_{Vib}$ controller;
- Case 3: a P-PI controller with velocity feedforward.

The experimental tracking error is shown in Figures 12 and 13. The maximum tracking error occurs in the acceleration and deceleration processes. In comparison with case 1 and case 3, the absolute value of maximum tracking error is reduced by about 20% and 40%, respectively, in case 2. Furthermore, as shown in Figure 12, the tracking error of case 1 is larger on both sides in the uniform process. This phenomenon may occur due to the system dynamics caused by motion changes. Case 2 significantly outperforms case 1, in which the tracking error is distributed near zero and remains almost stable in the uniform process. As shown in Figure 13, the tracking error of case 3 fluctuates considerably in the uniform process. As a result, case 2 has lower root-mean-square (RMS) error values than case 1 and case 3, as indicated in Table 2. In the case of structural vibration, the linear displacement and the rotary displacement can vary considerably in the working process. This displacement difference means that the screw is undertaking a large shaft torque and generates deformation. In this study, the error between linear and rotary displacement is used to characterize structural vibration, as shown in Figure 14. The vibration suppression performance of case 2 is better than that of case 1 and case 3, as demonstrated by the lower absolute maximum error and RMS error values reported in Table 2.

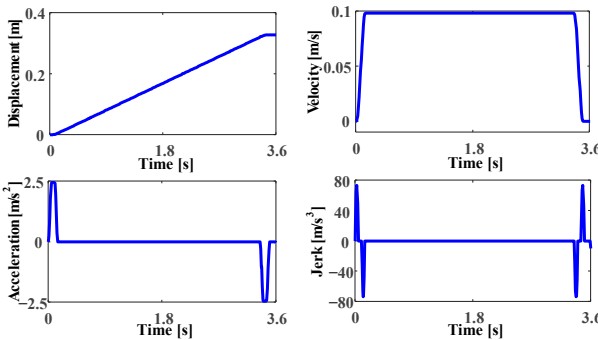

**Figure 11.** Reference trajectory.

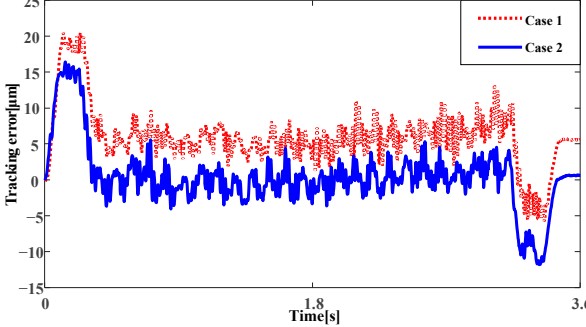

**Figure 12.** Experimental tracking error for case 1 (red dashed line) and for case 2 (blue solid line).

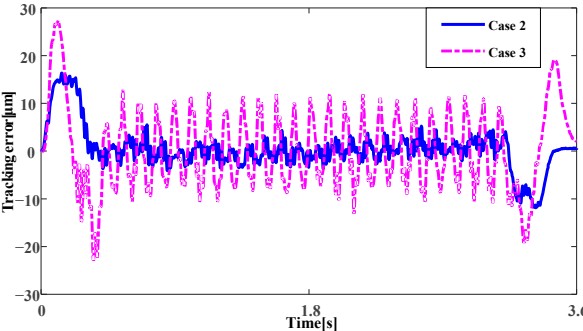

**Figure 13.** Experimental tracking error for case 2 (blue solid line) and for case 3 (pink dashed line).

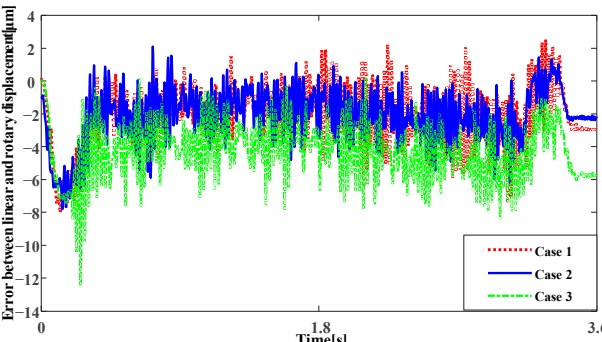

**Figure 14.** Error between linear and rotary displacement for case 1 (red dashed line), case 2 (blue solid line) and case 3 (green dashed line).

**Table 2.** Experimental tracking error values.

|  | Case | Max Abs. Error | RMS |
|---|---|---|---|
| Tracking error with no workpiece (μm) | 1 | 20.44 | 7.30 |
|  | 2 | 16.45 | 4.51 |
|  | 3 | 27.35 | 8.48 |
| Tracking error with a 25 kg workpiece added (μm) | 1 | 25.38 | 10.86 |
|  | 2 | 19.9 | 5.31 |
|  | 3 | 59.76 | 16.45 |
| Tracking error with harmonic disturbance (μm) | 1 | 27.70 | 9.28 |
|  | 2 | 17.60 | 5.22 |
|  | 3 | 41.15 | 10.96 |
| Error between linear and rotary displacement (μm) | 1 | 8.00 | 2.76 |
|  | 2 | 7.79 | 2.66 |
|  | 3 | 12.4 | 4.68 |

*6.5. Robust Performance Experiments for Mass Variation*

To verify the robust performance of the three controllers for mass variation, a 25 kg workpiece was added to the table, with the system following the trajectory shown in Figure 11. A comparison of the tracking performance with no workpiece and with a 25 kg workpiece is plotted in Figure 15 for three controllers. The maximum absolute and RMS errors with a 25 kg workpiece added are listed in Table 2. For case 2, the tracking performance is reduced by about 18% due to the adverse effect of mass variation. The tracking performance is reduced by about 49% and 94% for case 1 and case 3, respectively. These results demonstrate the robust performance of $K_{\text{Vib}}$ in case 2, reducing the adverse effect caused by load mass variation.

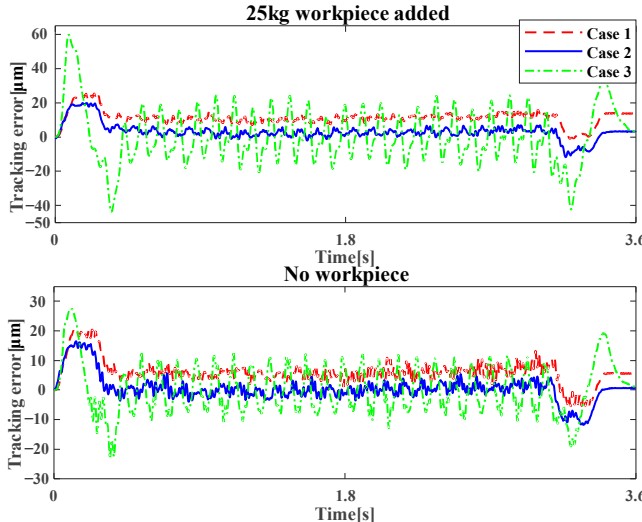

**Figure 15.** Tracking error of three controllers with no workpiece (blue solid line) and with a 25 kg workpiece added (red dashed line).

## 6.6. Vibration Suppression Performance Experiments

As an example to consider the effect of vibration excited by disturbance, a harmonic disturbance input voltage signal ($d = 1.2\sin(630t)$) V is applied to excite the flexible mode of the experimental setup, which is between 75 and 110 Hz (see in Figure 9), following the trajectory shown in Figure 11. The tracking error of three cases is plotted in Figure 16, and the maximum absolute error and RMS are listed in Table 2. For case 2, the tracking performance is reduced by about 16% due to the adverse effect of resonance. The tracking performance is reduced by about 27% and 29% for case 1 and case 3, respectively. Furthermore, for case 1 and case 3, the tracking error fluctuates more severely than for case 2. This result highlights the importance of vibration suppression in reducing the error in tracking control and avoiding resonance.

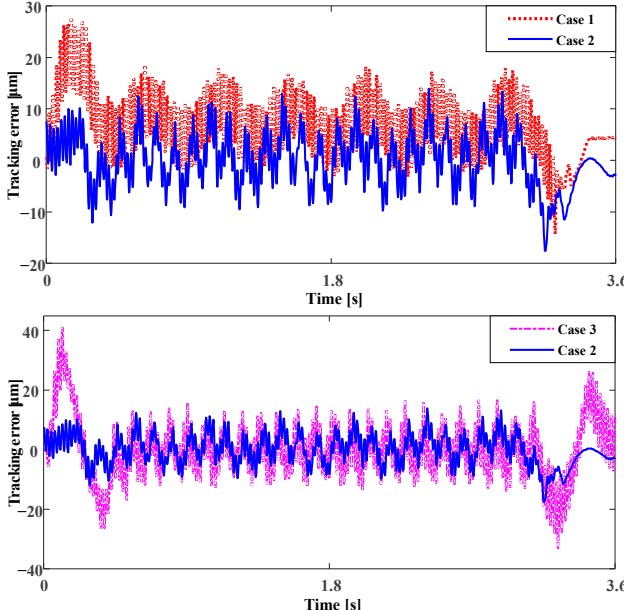

**Figure 16.** Tracking error of harmonic disturbance for case 1 (red dashed line), case 2 (red dashed line) and case 3 (pink dashed line).

*6.7. Disturbance-Rejection Performance Experiments*

In the uniform process with 0.1 m/s, a 5 kg sudden thrust was exerted on the table to simulate an external pulse disturbance. The experimental results are plotted in Figure 17. The maximum absolute tracking error in case 2 is 30.05 μm, whereas that in case 1 and case 3 is 36.05 μm and 29.7 μm, respectively. In addition, the setting time of case 2 is much shorter than that of case 1 and case 3, as shown in Figure 16, indicating that the proposed controller achieves the fastest response for disturbance rejection and highlighting the importance of $K_{Vib}$ in suppressing the influence of external disturbance. Therefore, case 2 has a better disturbance-rejection performance than case 1 and case 3.

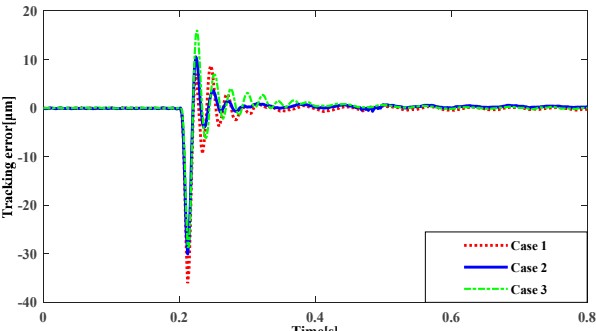

**Figure 17.** Tracking error with disturbance for case 1 (red dashed line), case 2 (blue solid line) and case 3 (green dashed line).

## 7. Conclusions

In this paper, a parallel structure controller was proposed to achieve high performance in ball screw drive systems for both tracking and vibration suppression. An LPV model for flexible ball screw drive systems was used to represent the dynamic variations caused by variations in the table position and workpiece mass. Then, an LQR controller and an interpolated gain-scheduled controller were designed to improve the tracking performance and damp out the dynamic flexible modes, respectively. The experimental results confirmed that the proposed controller achieves satisfactory performance for tracking, vibration suppression and disturbance rejection.

**Author Contributions:** Conceptualization, M.Z.; methodology, M.Z. and D.B.; software, M.Z.; validation, M.Z., D.B. and M.S.; formal analysis, M.Z.; writing—original draft preparation, M.Z.; writing—review and editing, M.Z. and Y.L.; funding acquisition, M.Z. All authors have read and agreed to the published version of the manuscript.

**Funding:** This research was funded by the Jiangsu Basic Research Program (Natural Science Foundation) Youth Fund Project Number BK20201046.

**Data Availability Statement:** Data sharing is not applicable to this article.

**Conflicts of Interest:** The authors declare no conflict of interest.

## Appendix A

Suppose that the stiffness ($\bar{k}_N$) and damping coefficient ($\bar{c}_N$) change within a certain range of $l_t$, $\bar{k}_N \in [k_i, k_{i+1}]$ and $\bar{c}_N \in [c_i, c_{i+1}]$; thus, the stiffness and damping coefficient can be expressed as

$$\bar{k}_N = k_N(1 + \widetilde{k}_N \delta_k), \ \bar{c}_N = c_N(1 + \widetilde{c}_N \delta_c) \tag{A1}$$

where $\delta_k(-1 \le \delta_k \le 1)$ and $\delta_c(-1 \le \delta_c \le 1)$ are the normalized real constant uncertainty of stiffness and the damping coefficient, respectively, and $\widetilde{k}_N$ and $\widetilde{c}_N$ represent the percentage of uncertainty of the stiffness and damping coefficient, respectively. The tracking controller

$K_{\text{Track}} = [k_1 \; k_2 \; k_3 \; k_4]$. Then, the system matrices of the plant in Equation (10) can be expressed as

$$
\widetilde{A} = \begin{bmatrix}
0 & 0 & 1 & 0 \\
0 & 0 & 0 & 1 \\
-\dfrac{k_N}{m_{2N}} & \dfrac{k_N}{m_{2N}} & -\dfrac{b_2+c_N}{m_{2N}} & \dfrac{c_N}{m_{2N}} \\
\dfrac{k_N-k_1}{m_{1N}} & -\dfrac{k_N+k_2}{m_{1N}} & \dfrac{c_N-k_3}{m_{1N}} & -\dfrac{b_1+c_N+k_4}{m_{1N}}
\end{bmatrix},
$$

$$
\widetilde{B}_1 = \begin{bmatrix}
0 & 0 \\
0 & 0 \\
-\dfrac{\widetilde{k}_N}{m_{2N}} & -\dfrac{\widetilde{c}_N}{m_{2N}} \\
\dfrac{\widetilde{k}_N}{m_{1N}} & \dfrac{\widetilde{c}_N}{m_{1N}}
\end{bmatrix},
$$

$$
\widetilde{B}_2 = \begin{bmatrix}
0 & 0 \\
0 & 0 \\
0 & \dfrac{1}{m_{2N}} \\
\dfrac{1}{m_{1N}} & 0
\end{bmatrix}, \widetilde{C}_1 = \begin{bmatrix}
k_N & -k_N & 0 & 0 \\
0 & 0 & c_N & -c_N
\end{bmatrix},
$$

$$
\widetilde{D}_{11} = \begin{bmatrix} 0 & 0 \\ 0 & 0 \end{bmatrix}, \widetilde{D}_{12} = \begin{bmatrix} 0 & 0 \\ 0 & 0 \end{bmatrix}
$$

$$
\widetilde{C}_2 = \begin{bmatrix} 1 & 0 & 0 & 0 \\ 0 & 1 & 0 & 0 \end{bmatrix}, \widetilde{D}_{21} = \begin{bmatrix} 0 & 0 \\ 0 & 0 \end{bmatrix}, \widetilde{D}_{22} = \begin{bmatrix} 0 & 0 \\ 0 & 0 \end{bmatrix}
$$

(A2)

## Appendix B

The $A_\mu$, $F_\mu$, $X_\mu$ and $Z_\mu$ matrices can be defined as follows [40].

$X_\mu$ and $Z_\mu$ are bounded, non-negative definite matrices according to the coupled Riccati equations:

$$
-\dot{X}_\mu = X_\mu P_{11} + P_{11}^{\mathrm{T}} X_\mu + X_\mu (P_{12}P_{12}^{\mathrm{T}}/\gamma^2 - P_{13}P_{13}^{\mathrm{T}})X_\mu + P_{21}^{\mathrm{T}} P_{21} \tag{A3}
$$

$$
\dot{Z}_\mu = Z_\mu P_{11} + Z_\mu P_{11}^{\mathrm{T}} + Z_\mu (X_\mu P_{13}P_{13}^{\mathrm{T}} X_\mu/\gamma^2 - P_{31}P_{31}^{\mathrm{T}})Z_\mu + P_{12}^{\mathrm{T}} P_{12} \tag{A4}
$$

$$
A_\mu = P_{11} + P_{13}F_\mu + \gamma^{-2}P_{12}P_{12}^{\mathrm{T}} X_\mu + N_\mu L_\mu P_{31},
$$
$$
F_\mu = -P_{12}^{\mathrm{T}} X_\mu, L_\mu = -Y_\mu P_{31}^{\mathrm{T}},
$$
$$
Y_\mu = (Z_\mu^{-1} + X_\mu) - 1, N_\mu = (I - Y_\mu X_\mu) - 1 \tag{A5}
$$

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
