# Peer review of "Tracking and Vibration Control with a Parallel Structure Controller Based on a Flexible Ball Screw Drive System"

_actuators, doi:10.3390/act12080330_

Round 1

Reviewer 1 Report

The paper is overly complicated for the ultimate goal. At the very least the system should be modeled using first principles. The experiment is basically a ball screw moving a mass. The authors should have recognized that the current and torque constants are the same value if they are expressed in SI units. This shows a lack of understanding of the physics of their system.

It looks like the introduction was written using AI. There are many words that are inappropriately hyphenated. This needs to be fixed.

There are countless trajectory tracking algorithms and many ways to perform vibration control. It is not convincing that the presentation here is the best way to accomplish their stated goals. The authors should compare their approach to alternative methods and clearly show that their approach is better or essential for this application.  

See comments to the authors.

Reviewer 2 Report

The article presented parallel structure controller for ball screw drive systems with tracking and vibration suppression capabilities. In general, the work is interesting and is evidence based with experimentations. However, the authors must address and incorporate the following comments to make the article suitable for publications in Actuators Journal.

1-      Remove the hyphen from the continuous words such as "ac-curacy", "exter-nal", “posi-tion”, etc.

2-      The authors should add details on how the values of the parameters for two parallel controller were realized from the physical system for control purposes.

3-      The authors mentioned on line 155-156 that "According to experimental results, b1 and b2 are being kept constant", However, I cannot find any reference to the experimental results.

4-      One lines 174 - 176, the authors mentioned "his section may be divided by subheadings. It should provide a concise and precise description of the experimental results, their interpretation, as well as the experimental conclusions that can be drawn. " which does not make sense and seems to be the comments from the professor for improvement. Including this in the paper seems unprofessional approach from the authors.

5-      For the vibration suppression the authors have adopted an LTI control model, while the stiffness, damping of ball screw, and the table location are time-varying. How the authors employed LTI control model to control the time-varying phenomenon? Add a detail description to the revised manuscript.

6-      Add a comparison of the proposed system with similar approaches from the published literature both quantitively and qualitatively.

Minor editing of English language required

Reviewer 3 Report

row 45: "resent" -> recent

I suggest to avoid the introduction of multiple resonances in (2) since only one resonance is considered for vibration suppression.

Please enlarge figues.

row 273: "vicious" -> viscous

I wonder if the poor performance of the simple LQR control is due to the lack of an integral action, not considered in the figure of merit.

Please define the input and output of the transfer function shown in Fig. 9.

The same request for Fig. 10, are the shown trnasfer functions computed with the LQR control closed?

The claim "The FRF of LTI control model combined by experimental setup LTI model with uncertainties in the position l [170,220]mm and tracking controller KTrack is displayed in Figure 10" is not clear.

English is good, only minor typos checking is needed.

Reviewer 4 Report

Feedback control strategies are implemented and tested experimentally for reducing disturbances in ball screw drives. Comparison is made against PI control. It is good to see experimental results and they seem convincing. The presentation in general could be improved. The language could benefit from a general review. There are a lot of hyphen symbols appearing in words where there should not be and it is not always clear what is being said.

It would be good to bring out the significance of the proposed control - why is it better than other approaches? In particular, the disturbance is likely harmonic at structural resonant frequencies in may cases. This type of disturbance would be far more effectively reduced with a periodic controller (e.g. LMS or many others, including those able to estimate the unknown frequency of the disturbance if it is subject to variation. These approaches should be included in the literature review and discussion.

Results are generally clear, but it may be better for fig 15 to present the three cases on the same axes as is done for other figures. i.e. have two figures - one with and one without the 25kg workpiece, each of the two figures showing three traces for the three controllers.

It is not easy to follow and would benefit from a thorough review.

Round 2

Reviewer 2 Report

The authors have addressed and incorporated all my comments in the revised manuscript which has improved the overall quality of the manuscript.